# A Residual Bootstrap for High-Dimensional Regression with Near Low-Rank Designs

**Miles E. Lopes**
Department of Statistics
University of California, Berkeley
Berkeley, CA 94720
mlopes@stat.berkeley.edu

## Abstract

We study the residual bootstrap (RB) method in the context of high-dimensional linear regression. Specifically, we analyze the distributional approximation of linear contrasts $c^\top(\widehat{\beta}_\rho - \beta)$, where $\widehat{\beta}_\rho$ is a ridge-regression estimator. When regression coefficients are estimated via least squares, classical results show that RB consistently approximates the laws of contrasts, provided that $p \ll n$, where the design matrix is of size $n \times p$. Up to now, relatively little work has considered how additional structure in the linear model may extend the validity of RB to the setting where $p/n \asymp 1$. In this setting, we propose a version of RB that resamples residuals obtained from ridge regression. Our main structural assumption on the design matrix is that it is nearly low rank — in the sense that its singular values decay according to a power-law profile. Under a few extra technical assumptions, we derive a simple criterion for ensuring that RB consistently approximates the law of a given contrast. We then specialize this result to study confidence intervals for mean response values $X_i^\top \beta$, where $X_i^\top$ is the $i$th row of the design. More precisely, we show that conditionally on a Gaussian design with near low-rank structure, RB *simultaneously* approximates all of the laws $X_i^\top(\widehat{\beta}_\rho - \beta)$, $i = 1, \ldots, n$. This result is also notable as it imposes no sparsity assumptions on $\beta$. Furthermore, since our consistency results are formulated in terms of the Mallows (Kantorovich) metric, the existence of a limiting distribution is not required.

## 1 Introduction

Until recently, much of the emphasis in the theory of high-dimensional statistics has been on "first order" problems, such as estimation and prediction. As the understanding of these problems has become more complete, attention has begun to shift increasingly towards "second order" problems, dealing with hypothesis tests, confidence intervals, and uncertainty quantification [1–6]. In this direction, much less is understood about the effects of structure, regularization, and dimensionality — leaving many questions open. One collection of such questions that has attracted growing interest deals with the operating characteristics of the *bootstrap* in high dimensions [7–9] . Due to the fact that bootstrap is among the most widely used tools for approximating the sampling distributions of test statistics and estimators, there is much practical value in understanding what factors allow for the bootstrap to succeed in the high-dimensional regime.

**The regression model and linear contrasts.** In this paper, we focus our attention on high-dimensional linear regression, and our aim is to know when the residual bootstrap (RB) method consistently approximates the laws of *linear contrasts*. (A review of RB is given in Section 2.)

To specify the model, suppose that we observe a response vector $Y \in \mathbb{R}^n$, generated according to

$$Y = X\beta + \varepsilon, \tag{1}$$

where $X \in \mathbb{R}^{n \times p}$ is the observed design matrix, $\beta \in \mathbb{R}^p$ is an unknown vector of coefficients, and the error variables $\varepsilon = (\varepsilon_1, \ldots, \varepsilon_n)$ are drawn i.i.d. from an unknown distribution $F_0$, with mean 0 and unknown variance $\sigma^2 < \infty$. As is conventional in high-dimensional statistics, we assume the model (1) is embedded in a sequence of models indexed by $n$. Hence, we allow $X$, $\beta$, and $p$ to vary implicitly with $n$. We will leave $p/n$ unconstrained until Section 3.3, where we will assume $p/n \asymp 1$ in Theorem 3, and then in Section 3.4, we will assume further that $p/n$ is bounded strictly between 0 and 1. The distribution $F_0$ is fixed with respect to $n$, and none of our results require $F_0$ to have more than four moments.

Although we are primarily interested in cases where the design matrix $X$ is deterministic, we will also study the performance of the bootstrap conditionally on a Gaussian design. For this reason, we will use the symbol $\mathbb{E}[\ldots | X]$ even when the design is non-random so that confusion does not arise in relating different sections of the paper. Likewise, the symbol $\mathbb{E}[\ldots]$ refers to unconditional expectation over all sources of randomness. Whenever the design is random, we will assume $X \perp\!\!\!\perp \varepsilon$, denoting the distribution of $X$ by $\mathbb{P}_X$, and the distribution of $\varepsilon$ by $\mathbb{P}_\varepsilon$.

Within the context of the regression, we will be focused on linear contrasts $c^\top(\widehat{\beta} - \beta)$, where $c \in \mathbb{R}^p$ is a fixed vector and $\widehat{\beta} \in \mathbb{R}^p$ is an estimate of $\beta$. The importance of contrasts arises from the fact that they unify many questions about a linear model. For instance, testing the significance of the $i$th coefficient $\beta_i$ may be addressed by choosing $c$ to be the standard basis vector $c^\top = e_i^\top$. Another important problem is quantifying the uncertainty of point predictions, which may be addressed by choosing $c^\top = X_i^\top$, i.e. the $i$th row of the design matrix. In this case, an approximation to the law of the contrast leads to a confidence interval for the mean response value $\mathbb{E}[Y_i] = X_i^\top\beta$. Further applications of contrasts occur in the broad topic of ANOVA [10].

**Intuition for structure and regularization in RB.** The following two paragraphs explain the core conceptual aspects of the paper. To understand the role of regularization in applying RB to high-dimensional regression, it is helpful to think of RB in terms of two ideas. First, if $\widehat{\beta}_{\mathrm{LS}}$ denotes the ordinary least squares estimator, then it is a simple but important fact that contrasts can be written as $c^\top(\widehat{\beta}_{\mathrm{LS}} - \beta) = a^\top\varepsilon$ where $a^\top := c^\top(X^\top X)^{-1}X^\top$. Hence, if it were possible to sample directly from $F_0$, then the law of any such contrast could be easily determined. Since $F_0$ is unknown, the second key idea is to use the residuals of *some* estimator $\widehat{\beta}$ as a proxy for samples from $F_0$. When $p \ll n$, the least-squares residuals are a good proxy [11, 12]. However, it is well-known that least-squares tends to overfit when $p/n \asymp 1$. When $\widehat{\beta}_{\mathrm{LS}}$ fits "too well", this means that its residuals are "too small", and hence they give a poor proxy for $F_0$. Therefore, by using a regularized estimator $\widehat{\beta}$, overfitting can be avoided, and the residuals of $\widehat{\beta}$ may offer a better way of obtaining "approximate samples" from $F_0$.

The form of regularized regression we will focus on is *ridge regression*:

$$\widehat{\beta}_\rho := (X^\top X + \rho I_{p \times p})^{-1}X^\top Y, \tag{2}$$

where $\rho > 0$ is a user-specified regularization parameter. As will be seen in Sections 3.2 and 3.3, the residuals obtained from ridge regression lead to a particularly good approximation of $F_0$ when the design matrix $X$ is nearly low-rank, in the sense that most of its singular values are close to 0. In essence, this condition is a form of sparsity, since it implies that the rows of $X$ nearly lie in a low-dimensional subspace of $\mathbb{R}^p$. However, this type of structural condition has a significant advantage over the the more well-studied assumption that $\beta$ is sparse. Namely, the assumption that $X$ is nearly low-rank can be inspected directly in practice — whereas sparsity in $\beta$ is typically unverifiable. In fact, our results will impose no conditions on $\beta$, other than that $\|\beta\|_2$ remains bounded as $(n, p) \to \infty$. Finally, it is worth noting that the occurrence of near low-rank design matrices is actually very common in applications, and is often referred to as *collinearity* [13, ch. 17].

**Contributions and outline.** The primary contribution of this paper is a complement to the work of Bickel and Freedman [12] (hereafter B&F 1983) — who showed that in general, the RB method fails

to approximate the laws of least-squares contrasts $c^\top(\widehat{\beta}_{\mathrm{LS}} - \beta)$ when $p/n \asymp 1$. Instead, we develop an alternative set of results, proving that even when $p/n \asymp 1$, RB can successfully approximate the laws of "ridged contrasts" $c^\top(\widehat{\beta}_\rho - \beta)$ for many choices of $c \in \mathbb{R}^p$, provided that the design matrix $X$ is nearly low rank. A particularly interesting consequence of our work is that RB successfully approximates the law $c^\top(\widehat{\beta}_\rho - \beta)$ for a certain choice of $c$ that was shown in B&F 1983 to "break" RB when applied to least-squares. Specifically, such a $c$ can be chosen as one of the rows of $X$ with a high *leverage score* (see Section 4). This example corresponds to the practical problem of setting confidence intervals for mean response values $\mathbb{E}[Y_i] = X_i^\top \beta$. (See [12, p. 41], as well as Lemma 2 and Theorem 4 in Section 3.4). Lastly, from a technical point of view, a third notable aspect of our results is that they are formulated in terms of the Mallows-$\ell_2$ metric, which frees us from having to impose conditions that force a limiting distribution to exist.

Apart from B&F 1983, the most closely related works we are aware of are the recent papers [7] and [8], which also consider RB in the high-dimensional setting. However, these works focus on role of sparsity in $\beta$ and do not make use of low-rank structure in the design, whereas our work deals only with structure in the design and imposes no sparsity assumptions on $\beta$.

The remainder of the paper is organized as follows. In Section 2, we formulate the problem of approximating the laws of contrasts, and describe our proposed methodology for RB based on ridge regression. Then, in Section 3 we state several results that lay the groundwork for Theorem 4, which shows that that RB can successfully approximate all of the laws $\mathcal{L}(X_i^\top(\widehat{\beta}_\rho - \beta)|X)$, $i = 1, \ldots, n$, conditionally on a Gaussian design. Due to space constraints, all proofs are deferred to material that will appear in a separate work.

**Notation and conventions.** If $U$ and $V$ are random variables, then $\mathcal{L}(U|V)$ denotes the law of $U$, conditionally on $V$. If $a_n$ and $b_n$ are two sequences of real numbers, then the notation $a_n \lesssim b_n$ means that there is an absolute constant $\kappa_0 > 0$ and an integer $n_0 \geq 1$ such that $a_n \leq \kappa_0 b_n$ for all $n \geq n_0$. The notation $a_n \asymp b_n$ means that $a_n \lesssim b_n$ and $b_n \lesssim a_n$. For a square matrix $A \in \mathbb{R}^{k \times k}$ whose eigenvalues are real, we denote them by $\lambda_{\min}(A) = \lambda_k(A) \leq \cdots \leq \lambda_1(A) = \lambda_{\max}(A)$.

## 2 Problem setup and methodology

**Problem setup.** For any $c \in \mathbb{R}^p$, it is clear that conditionally on $X$, the law of $c^\top(\widehat{\beta}_\rho - \beta)$ is completely determined by $F_0$, and hence it makes sense to use the notation

$$\Psi_\rho(F_0; c) := \mathcal{L}\big(c^\top(\widehat{\beta}_\rho - \beta) \,\big|\, X\big). \tag{3}$$

The problem we aim to solve is to approximate the distribution $\Psi_\rho(F_0; c)$ for suitable choices of $c$.

**Review of the residual bootstrap (RB) procedure.** We briefly explain the steps involved in the residual bootstrap procedure, applied to the ridge estimator $\widehat{\beta}_\rho$ of $\beta$. To proceed somewhat indirectly, consider the following "bias-variance" decomposition of $\Psi_\rho(F_0; c)$, conditionally on $X$,

$$\Psi_\rho(F_0; c) = \underbrace{\mathcal{L}\big(c^\top\big(\widehat{\beta}_\rho - \mathbb{E}[\widehat{\beta}_\rho|X]\big) \,\big|\, X\big)}_{=:\, \Phi_\rho(F_0; c)} + \underbrace{c^\top\big(\mathbb{E}[\widehat{\beta}_\rho|X] - \beta\big)}_{=:\, \mathrm{bias}(\Phi_\rho(F_0; c))}. \tag{4}$$

Note that the distribution $\Phi(F_0; c)$ has mean zero, and so that the second term on the right side is the bias of $\Phi_\rho(F_0; c)$ as an estimator of $\Psi_\rho(F_0; c)$. Furthermore, the distribution $\Phi_\rho(F_0; c)$ may be viewed as the "variance component" of $\Psi_\rho(F_0; c)$. We will be interested in situations where the regularization parameter $\rho$ may be chosen small enough so that the bias component is small. In this case, one has $\Psi_\rho(F_0; c) \approx \Phi_\rho(F_0; c)$, and then it is enough to find an approximation to the law $\Phi_\rho(F_0; c)$, which is unknown. To this end, a simple manipulation of $c^\top(\widehat{\beta}_\rho - \mathbb{E}[\widehat{\beta}_\rho])$ leads to

$$\Phi_\rho(F_0; c) = \mathcal{L}(c^\top(X^\top X + \rho I_{p \times p})^{-1} X^\top \varepsilon \,\big|\, X). \tag{5}$$

Now, to approximate $\Phi_\rho(F_0; c)$, let $\widehat{F}$ be any centered estimate of $F_0$. (Typically, $\widehat{F}$ is obtained by using the centered residuals of some estimator of $\beta$, but this is not necessary in general.) Also, let $\varepsilon^* = (\varepsilon_1^*, \ldots, \varepsilon_n^*) \in \mathbb{R}^n$ be an i.i.d. sample from $\widehat{F}$. Then, replacing $\varepsilon$ with $\varepsilon^*$ in line (5) yields

$$\Phi_\rho(\widehat{F}; c) = \mathcal{L}(c^\top(X^\top X + \rho I_{p \times p})^{-1} X^\top \varepsilon^* \,\big|\, X). \tag{6}$$

At this point, we define the (random) measure $\Phi_\rho(\widehat{F}; c)$ to be the RB approximation to $\Phi_\rho(F_0; c)$. Hence, it is clear that the RB approximation is simply a "plug-in rule".

**A two-stage approach.** An important feature of the procedure just described is that we are free to use any centered estimator $\widehat{F}$ of $F_0$. This fact offers substantial flexibility in approximating $\Psi_\rho(F_0; c)$. One way of exploiting this flexibility is to consider a two-stage approach, where a "pilot" ridge estimator $\widehat{\beta}_\varrho$ is used to first compute residuals whose centered empirical distribution function is $\widehat{F}_\varrho$, say. Then, in the second stage, the distribution $\widehat{F}_\varrho$ is used to approximate $\Phi_\rho(F_0; c)$ via the relation (6). To be more detailed, if $(\widehat{e}_1(\varrho), \ldots, \widehat{e}_n(\varrho)) = \widehat{e}(\varrho) := Y - X\widehat{\beta}_\varrho$ are the residuals of $\widehat{\beta}_\varrho$, then we define $\widehat{F}_\varrho$ to be the distribution that places mass $1/n$ at each of the values $\widehat{e}_i(\varrho) - \bar{e}(\varrho)$ with $\bar{e}(\varrho) := \frac{1}{n} \sum_{i=1}^n \widehat{e}_i(\varrho)$. Here, it is important to note that the value $\varrho$ is chosen to optimize $\widehat{F}_\varrho$ as an approximation to $F_0$. By contrast, the choice of $\rho$ depends on the relative importance of width and coverage probability for confidence intervals based on $\Phi_\rho(\widehat{F}_\varrho; c)$. Theorems 1, 3, and 4 will offer some guidance in selecting $\varrho$ and $\rho$.

**Resampling algorithm.** To summarize the discussion above, if $B$ is user-specified number of bootstrap replicates, our proposed method for approximating $\Psi_\rho(F_0; c)$ is given below.

1. Select $\rho$ and $\varrho$, and compute the residuals $\widehat{e}(\varrho) = Y - X\widehat{\beta}_\varrho$.

2. Compute the centered distribution function $\widehat{F}_\varrho$, putting mass $1/n$ at each $\widehat{e}_i(\varrho) - \bar{e}(\varrho)$.

3. For $j = 1, \ldots, B$:
   - Draw a vector $\varepsilon^* \in \mathbb{R}^n$ of $n$ i.i.d. samples from $\widehat{F}_\varrho$.
   - Compute $z_j := c^\top (X^\top X + \rho I_{p \times p})^{-1} X^\top \varepsilon^*$.

4. Return the empirical distribution of $z_1, \ldots, z_B$.

Clearly, as $B \to \infty$, the empirical distribution of $z_1, \ldots, z_B$ converges weakly to $\Phi_\rho(\widehat{F}_\varrho; c)$, with probability 1. As is conventional, our theoretical analysis in the next section will ignore Monte Carlo issues, and address only the performance of $\Phi_\rho(\widehat{F}_\varrho; c)$ as an approximation to $\Psi_\rho(F_0; c)$.

## 3 Main results

The following metric will be central to our theoretical results, and has been a standard tool in the analysis of the bootstrap, beginning with the work of Bickel and Freedman [14].

**The Mallows (Kantorovich) metric.** For two random vectors $U$ and $V$ in a Euclidean space, the Mallows-$\ell_2$ metric is defined by

$$d_2^2(\mathcal{L}(U), \mathcal{L}(V)) := \inf_{\pi \in \Pi} \left\{ \mathbb{E}\left[ \|U - V\|_2^2 \right] : (U, V) \sim \pi \right\} \tag{7}$$

where the infimum is over the class $\Pi$ of joint distributions $\pi$ whose marginals are $\mathcal{L}(U)$ and $\mathcal{L}(V)$. It is worth noting that convergence in $d_2$ is strictly stronger than weak convergence, since it also requires convergence of second moments. Additional details may be found in the paper [14].

### 3.1 A bias-variance decomposition for bootstrap approximation

To give some notation for analyzing the bias-variance decomposition of $\Psi_\rho(F_0; c)$ in line (4), we define the following quantities based upon the ridge estimator $\widehat{\beta}_\rho$. Namely, the variance is

$$v_\rho = v_\rho(X; c) := \operatorname{var}(\Psi_\rho(F_0; c)|X) = \sigma^2 \|c^\top (X^\top X + \rho I_{p \times p})^{-1} X^\top\|_2^2.$$

To express the bias of $\Phi_\rho(F_0; c)$, we define the vector $\delta(X) \in \mathbb{R}^p$ according to

$$\delta(X) := \beta - \mathbb{E}[\widehat{\beta}_\rho] = \left[ I_{p \times p} - (X^\top X + \rho I_{p \times p})^{-1} X^\top X \right]\beta, \tag{8}$$

and then put

$$b_\rho^2 = b_\rho^2(X; c) := \text{bias}^2(\Phi_\rho(F_0; c)) = (c^\top \delta(X))^2. \qquad (9)$$

We will sometimes omit the arguments of $v_\rho$ and $b_\rho^2$ to lighten notation. Note that $v_\rho(X; c)$ does not depend on $\beta$, and $b_\rho^2(X; c)$ only depends on $\beta$ through $\delta(X)$.

The following result gives a regularized and high-dimensional extension of some lemmas in Freedman's early work [11] on RB for least squares. The result does not require any structural conditions on the design matrix, or on the true parameter $\beta$.

**Theorem 1** (consistency criterion). *Suppose $X \in \mathbb{R}^{n \times p}$ is fixed. Let $\widehat{F}$ be any estimator of $F_0$, and let $c \in \mathbb{R}^p$ be any vector such that $v_\rho = v_\rho(X; c) \neq 0$. Then with $\mathbb{P}_\varepsilon$-probability 1, the following inequality holds for every $n \geq 1$, and every $\rho > 0$,*

$$d_2^2\left(\tfrac{1}{\sqrt{v_\rho}}\Psi_\rho(F_0; c), \tfrac{1}{\sqrt{v_\rho}}\Phi_\rho(\widehat{F}; c)\right) \leq \tfrac{1}{\sigma^2}d_2^2(F_0, \widehat{F}) + \tfrac{b_\rho^2}{v_\rho}. \qquad (10)$$

**Remarks.** Observe that the normalization $1/\sqrt{v_\rho}$ ensures that the bound is non-trivial, since the distribution $\Psi_\rho(F_0; c)/\sqrt{v_\rho}$ has variance equal to 1 for all $n$ (and hence does not become degenerate for large $n$). To consider the choice of $\rho$, it is simple to verify that the ratio $b_\rho^2/v_\rho$ decreases monotonically as $\rho$ decreases. Note also that as $\rho$ becomes small, the variance $v_\rho$ becomes large, and likewise, confidence intervals based on $\Phi_\rho(\widehat{F}; c)$ become wider. In other words, there is a trade-off between the width of the confidence interval and the size of the bound (10).

**Sufficient conditions for consistency of RB.** An important practical aspect of Theorem 1 is that for any given contrast $c$, the variance $v_\rho(X; c)$ can be easily estimated, since it only requires an estimate of $\sigma^2$, which can be obtained from $\widehat{F}$. Consequently, whenever theoretical bounds on $d_2^2(F_0, \widehat{F})$ and $b_\rho^2(X; c)$ are available, the right side of line (10) can be controlled. In this way, Theorem 1 offers a simple route for guaranteeing that RB is consistent. In Sections 3.2 and 3.3 to follow, we derive a bound on $\mathbb{E}[d_2^2(F_0, \widehat{F})|X]$ in the case where $\widehat{F}$ is chosen to be $\widehat{F}_\varrho$. Later on in Section 3.4, we study RB consistency in the context of prediction with a Gaussian design, and there we derive high probability bounds on both $v_\rho(X; c)$ and $b_\rho^2(X; c)$ where $c$ is a particular row of $X$.

## 3.2 A link between bootstrap consistency and MSPE

If $\widehat{\beta}$ is an estimator of $\beta$, its mean-squared prediction error (MSPE), conditionally on $X$, is defined as

$$\text{mspe}(\widehat{\beta}\,|X) := \tfrac{1}{n}\mathbb{E}\big[\|X(\widehat{\beta} - \beta)\|_2^2 \,\big|\, X\big]. \qquad (11)$$

The previous subsection showed that in-law approximation of contrasts is closely tied to the approximation of $F_0$. We now take a second step of showing that if the centered residuals of an estimator $\widehat{\beta}$ are used to approximate $F_0$, then the quality of this approximation can be bounded naturally in terms of $\text{mspe}(\widehat{\beta}\,|X)$. This result applies to any estimator $\widehat{\beta}$ computed from the observations (1).

**Theorem 2.** *Suppose $X \in \mathbb{R}^{n \times p}$ is fixed. Let $\widehat{\beta}$ be any estimator of $\beta$, and let $\widehat{F}$ be the empirical distribution of the centered residuals of $\widehat{\beta}$. Also, let $F_n$ denote the empirical distribution of $n$ i.i.d. samples from $F_0$. Then for every $n \geq 1$,*

$$\mathbb{E}\big[d_2^2(\widehat{F}, F_0) \,\big|\, X\big] \leq 2\,\text{mspe}(\widehat{\beta}\,|X) + 2\,\mathbb{E}[d_2^2(F_n, F_0)] + \tfrac{2\sigma^2}{n}. \qquad (12)$$

**Remarks.** As we will see in the next section, the MSPE of ridge regression can be bounded in a sharp way when the design matrix is approximately low rank, and there we will analyze $\text{mspe}(\widehat{\beta}_\varrho|X)$ for the pilot estimator. Consequently, when near low-rank structure is available, the only remaining issue in controlling the right side of line (12) is to bound the quantity $\mathbb{E}[d_2^2(F_n, F_0)|X]$. The very recent work of Bobkov and Ledoux [15] provides an in-depth study of this question, and they derive a variety bounds under different tail conditions on $F_0$. We summarize one of their results below.

**Lemma 1** (Bobkov and Ledoux, 2014). *If $F_0$ has a finite fourth moment, then*

$$\mathbb{E}[d_2^2(F_n, F_0)] \lesssim \log(n)n^{-1/2}. \qquad (13)$$

**Remarks.** The fact that the *squared* distance is bounded at the rate of $\log(n)n^{-1/2}$ is an indication that $d_2$ is a rather strong metric on distributions. For a detailed discussion of this result, see Corollaries 7.17 and 7.18 in the paper [15]. Although it is possible to obtain faster rates when more stringent tail conditions are placed on $F_0$, we will only need a fourth moment, since the $\mathrm{mspe}(\widehat{\beta}|X)$ term in Theorem 2 will often have a slower rate than $\log(n)n^{-1/2}$, as discussed in the next section.

### 3.3 Consistency of ridge regression in MSPE for near low rank designs

In this subsection, we show that when the tuning parameter $\varrho$ is set at a suitable rate, the pilot ridge estimator $\widehat{\beta}_\varrho$ is consistent in MSPE when the design matrix is near low-rank — even when $p/n$ is large, and without any sparsity constraints on $\beta$. We now state some assumptions.

**A1.** *There is a number $\nu > 0$, and absolute constants $\kappa_1, \kappa_2 > 0$, such that*

$$\kappa_1 i^{-\nu} \leq \lambda_i(\widehat{\Sigma}) \leq \kappa_2 i^{-\nu} \qquad for\ all \quad i = 1, \ldots, n \wedge p.$$

**A2.** *There are absolute constants $\theta, \gamma > 0$, such that for every $n \geq 1$, $\frac{\varrho}{n} = n^{-\theta}$ and $\frac{p}{n} = n^{-\gamma}$.*

**A3.** *The vector $\beta \in \mathbb{R}^p$ satisfies $\|\beta\|_2 \lesssim 1$.*

Due to Theorem 2, the following bound shows that the residuals of $\widehat{\beta}_\varrho$ may be used to extract a consistent approximation to $F_0$. Two other notable features of the bound are that it is *non-asymptotic* and *dimension-free*.

**Theorem 3.** *Suppose that $X \in \mathbb{R}^{n \times p}$ is fixed and that Assumptions 1–3 hold, with $p/n \asymp 1$. Assume further that $\theta$ is chosen as $\theta = \frac{2\nu}{3}$ when $\nu \in (0, \frac{1}{2})$, and $\theta = \frac{\nu}{\nu+1}$ when $\nu > \frac{1}{2}$. Then,*

$$\mathrm{mspe}(\widehat{\beta}_\varrho|X) \lesssim \left\{ \begin{array}{ll} n^{-\frac{2\nu}{3}} & if \quad \nu \in (0, \frac{1}{2}), \\ n^{-\frac{\nu}{\nu+1}} & if \quad \nu > \frac{1}{2}. \end{array} \right. \tag{14}$$

*Also, both bounds in (14) are tight in the sense that $\beta$ can be chosen so that $\widehat{\beta}_\varrho$ attains either rate.*

**Remarks.** Since the eigenvalues $\lambda_i(\widehat{\Sigma})$ are observable, they may be used to estimate $\nu$ and guide the selection of $\varrho/n = n^{-\theta}$. However, from a practical point of view, we found it easier to select $\varrho$ via cross-validation in numerical experiments, rather than via an estimate of $\nu$.

**A link with Pinsker's Theorem.** In the particular case when $F_0$ is a centered Gaussian distribution, the "prediction problem" of estimating $X\beta$ is very similar to estimating the mean parameters of a Gaussian sequence model, with error measured in the $\ell_2$ norm. In the alternative sequence-model format, the decay condition on the eigenvalues of $\frac{1}{n}X^\top X$ translates into an ellipsoid constraint on the mean parameter sequence [16, 17]. For this reason, Theorem 3 may be viewed as "regression version" of $\ell_2$ error bounds for the sequence model under an ellipsoid constraint (cf. Pinsker's Theorem, [16, 17]). Due to the fact that the latter problem has a very well developed literature, there may be various "neighboring results" elsewhere. Nevertheless, we could not find a direct reference for our stated MSPE bound in the current setup. For the purposes of our work in this paper, the more important point to take away from Theorem 3 is that it can be coupled with Theorem 2 for proving consistency of RB.

### 3.4 Confidence intervals for mean responses, conditionally on a Gaussian design

In this section, we consider the situation where the design matrix $X$ has rows $X_i^\top \in \mathbb{R}^p$ drawn i.i.d. from a multivariate normal distribution $N(0, \Sigma)$, with $X \perp\!\!\!\perp \varepsilon$. (The covariance matrix $\Sigma$ may vary with $n$.) Conditionally on a realization of $X$, we analyze the RB approximation of the laws $\Psi_\rho(F_0; X_i) = \mathcal{L}(X_i^\top(\widehat{\beta}_\rho - \beta)|X)$. As discussed in Section 1, this corresponds to the problem of setting confidence intervals for the mean responses $\mathbb{E}[Y_i] = X_i^\top \beta$. Assuming that the population eigenvalues $\lambda_i(\Sigma)$ obey a decay condition, we show below in Theorem 4 that RB succeeds with high $\mathbb{P}_X$-probability. Moreover, this consistency statement holds for all of the laws $\Psi_\rho(F_0; X_i)$ *simultaneously*. That is, among the $n$ distinct laws $\Psi_\rho(F_0; X_i)$, $i = 1, \ldots, n$, even the worst bootstrap approximation is still consistent. We now state some population-level assumptions.

**A4.** *The operator norm of $\Sigma \in \mathbb{R}^{p \times p}$ satisfies $\|\Sigma\|_{\text{op}} \lesssim 1$.*

Next, we impose a decay condition on the eigenvalues of $\Sigma$. This condition also ensures that $\Sigma$ is invertible for each fixed $p$ — even though the bottom eigenvalue may become arbitrarily small as $p$ becomes large. It is important to notice that we now use $\eta$ for the decay exponent of the population eigenvalues, whereas we used $\nu$ when describing the sample eigenvalues in the previous section.

**A5.** *There is a number $\eta > 0$, and absolute constants $k_1, k_2 > 0$, such that for all $i = 1, \ldots, p$,*

$$k_1 i^{-\eta} \leq \lambda_i(\Sigma) \leq k_2 i^{-\eta}.$$

**A6.** *There are absolute constants $k_3, k_4 \in (0,1)$ such that for all $n \geq 3$, we have the bounds $k_3 \leq \frac{p}{n} \leq k_4$ and $p \leq n - 2$.*

The following lemma collects most of the effort needed in proving our final result in Theorem 4. Here it is also helpful to recall the notation $\rho/n = n^{-\gamma}$ and $\varrho/n = n^{-\theta}$ from Assumption 2.

**Lemma 2.** *Suppose that the matrix $X \in \mathbb{R}^{n \times p}$ has rows $X_i^\top$ drawn i.i.d. from $N(0, \Sigma)$, and that Assumptions 2–6 hold. Furthermore, assume that $\gamma$ chosen so that $0 < \gamma < \min\{\eta, 1\}$. Then, the statements below are true.*

*(i) (bias inequality)*
*Fix any $\tau > 0$. Then, there is an absolute constant $\kappa_0 > 0$, such that for all large $n$, the following event holds with $\mathbb{P}_X$-probability at least $1 - n^{-\tau} - ne^{-n/16}$,*

$$\max_{1 \leq i \leq n} b_\rho^2(X; X_i) \leq \kappa_0 \cdot n^{-\gamma} \cdot (\tau + 1) \log(n + 2). \tag{15}$$

*(ii) (variance inequality)*
*There are absolute constants $\kappa_1, \kappa_2 > 0$ such that for all large $n$, the following event holds with $\mathbb{P}_X$-probability at least $1 - 4n \exp(-\kappa_1 n^{\frac{\gamma}{\eta}})$,*

$$\max_{1 \leq i \leq n} \frac{1}{v_\rho(X; X_i)} \leq \kappa_2 n^{1 - \frac{\gamma}{\eta}}. \tag{16}$$

*(iii) (mspe inequalities)*
*Suppose that $\theta$ is chosen as $\theta = 2\eta/3$ when $\eta \in (0, \frac{1}{2})$, and that $\theta$ is chosen as $\theta = \frac{\eta}{1+\eta}$ when $\eta > \frac{1}{2}$. Then, there are absolute constants $\kappa_3, \kappa_4, \kappa_5, \kappa_6 > 0$ such that for all large $n$,*

$$\text{mspe}(\widehat{\beta}_\varrho | X) \leq \begin{cases} \kappa_4 n^{-\frac{2\eta}{3}} & \text{with } \mathbb{P}_X\text{-probability at least } 1 - \exp(-\kappa_3 n^{2 - 4\eta/3}), & \text{if } \eta \in (0, \frac{1}{2}) \\ \kappa_6 n^{-\frac{\eta}{\eta+1}} & \text{with } \mathbb{P}_X\text{-probability at least } 1 - \exp(-\kappa_5 n^{\frac{2}{1+\eta}}), & \text{if } \eta > \frac{1}{2}. \end{cases}$$

**Remarks.** Note that the two rates in part (iii) coincide as $\eta$ approaches $1/2$. At a conceptual level, the entire lemma may be explained in relatively simple terms. Viewing the quantities $\text{mspe}(\widehat{\beta}_\varrho | X)$, $b_\rho^2(X; X_i)$ and $v_\rho(X; X_i)$ as functionals of a Gaussian matrix, the proof involves deriving concentration bounds for each of them. Indeed, this is plausible given that these quantities are smooth functionals of $X$. However, the difficulty of the proof arises from the fact that they are also highly non-linear functionals of $X$. We now combine Lemmas 1 and 2 with Theorems 1 and 2 to show that all of the laws $\Psi_\rho(F_0; X_i)$ can be simultaneously approximated via our two-stage RB method.

**Theorem 4.** *Suppose that $F_0$ has a finite fourth moment, Assumptions 2–6 hold, and $\gamma$ is chosen so that $\frac{\eta}{1+\eta} < \gamma < \min\{\eta, 1\}$. Also suppose that $\theta$ is chosen as $\theta = 2\eta/3$ when $\eta \in (0, \frac{1}{2})$, and $\theta = \frac{\eta}{\eta+1}$ when $\eta > \frac{1}{2}$. Then, there is a sequence of positive numbers $\delta_n$ with $\lim_{n \to \infty} \delta_n = 0$, such that the event*

$$\mathbb{E}\left[ \max_{1 \leq i \leq n} d_2^2\left( \frac{1}{\sqrt{v_\rho}} \Psi_\rho(F_0; X_i), \frac{1}{\sqrt{v_\rho}} \Phi_\rho(\widehat{F}_\varrho; X_i) \right) \Big| X \right] \leq \delta_n \tag{17}$$

*has $\mathbb{P}_X$-probability tending to 1 as $n \to \infty$.*

**Remark.** Lemma 2 gives explicit bounds on the numbers $\delta_n$, as well as the probabilities of the corresponding events, but we have stated the result in this way for the sake of readability.

## 4  Simulations

In four different settings of $n, p$, and the decay parameter $\eta$, we compared the nominal 90% confidence intervals (CIs) of four methods: "oracle", "ridge", "normal", and "OLS", to be described below. In each setting, we generated $N_1 := 100$ random designs $X$ with i.i.d. rows drawn from $N(0, \Sigma)$, where $\lambda_j(\Sigma) = j^{-\eta}$, $j = 1, \ldots, p$, and the eigenvectors of $\Sigma$ were drawn randomly by setting them to be the $Q$ factor in a $QR$ decomposition of a standard $p \times p$ Gaussian matrix. Then, for each realization of $X$, we generated $N_2 := 1000$ realizations of $Y$ according to the model (1), where $\beta = \mathbf{1}/\|\mathbf{1}\|_2 \in \mathbb{R}^p$, and $F_0$ is the centered $t$ distribution on 5 degrees of freedom, rescaled to have standard deviation $\sigma = 0.1$. For each $X$, and each corresponding $Y$, we considered the problem of setting a 90% CI for the mean response value $X_{i^\star}^\top \beta$, where $X_{i^\star}^\top$ is the row with the highest leverage score, i.e. $i^\star = \mathrm{argmax}_{1 \leq i \leq n} H_{ii}$ and $H := X(X^\top X)^{-1} X^\top$. This problem was shown in B&F 1983 to be a case where the standard RB method based on least-squares fails when $p/n \asymp 1$. Below, we refer to this method as "OLS".

To describe the other three methods, "ridge" refers to the interval $[X_{i^\star}^\top \widehat{\beta}_\rho - \widehat{q}_{0.95}, X_{i^\star}^\top \widehat{\beta}_\rho - \widehat{q}_{0.05}]$, where $\widehat{q}_\alpha$ is the $\alpha$% quantile of the numbers $z_1, \ldots, z_B$ computed in the proposed algorithm in Section 2, with $B = 1000$ and $c^\top = X_{i^\star}^\top$. To choose the parameters $\rho$ and $\varrho$ for a given $X$ and $Y$, we first computed $\widehat{r}$ as the value that optimized the MSPE error of a ridge estimator $\widehat{\beta}_r$ with respect to 5-fold cross validation; i.e. cross validation was performed for every distinct pair $(X, Y)$. We then put $\varrho = 5\widehat{r}$ and $\rho = 0.1\widehat{r}$, as we found the prefactors 5 and 0.1 to work adequately across various settings. (Optimizing $\varrho$ with respect to MSPE is motivated by Theorems 1, 2, and 3. Also, choosing $\rho$ to be somewhat smaller than $\varrho$ conforms with the constraints on $\theta$ and $\gamma$ in Theorem 4.) The method "normal" refers to the CI based on the normal approximation $\mathcal{L}(X_{i^\star}^\top (\widehat{\beta}_\rho - \beta)|X) \approx N(0, \widehat{\tau}^2)$, where $\widehat{\tau}^2 = \widehat{\sigma}^2 \|X_{i^\star}^\top (X^\top X + \rho I_{p \times p})^{-1} X^\top\|_2^2$, $\rho = 0.1\widehat{r}$, and $\widehat{\sigma}^2$ is the usual unbiased estimate of $\sigma^2$ based on OLS residuals. The "oracle" method refers to the interval $[X_{i^\star}^\top \widehat{\beta}_\rho - \tilde{q}_{0.95}, X_{i^\star}^\top \widehat{\beta}_\rho - \tilde{q}_{0.05}]$, with $\rho = 0.1\widehat{r}$, and $\tilde{q}_\alpha$ being the empirical $\alpha$% quantile of $X_i^\top(\widehat{\beta}_\rho - \beta)$ over all 1000 realizations of $Y$ based on a given $X$. (This accounts for the randomness in $\rho = 0.1\widehat{r}$.)

Within a given setting of the triplet $(n, p, \eta)$, we refer to the "coverage" of a method as the fraction of the $N_1 \times N_2 = 10^5$ instances where the method's CI contained the parameter $X_{i^\star}^\top \beta$. Also, we refer to "width" as the average width of a method's intervals over all of the $10^5$ instances. The four settings of $(n, p, \eta)$ correspond to moderate/high dimension and moderate/fast decay of the eigenvalues $\lambda_i(\Sigma)$. Even in the moderate case of $p/n = 0.45$, the results show that the OLS intervals are too narrow and have coverage noticeably less than 90%. As expected, this effect becomes more pronounced when $p/n = 0.95$. The ridge and normal intervals perform reasonably well across settings, with both performing much better than OLS. However, it should be emphasized that our study of RB is motivated by the desire to gain insight into the behavior of the bootstrap in high dimensions — rather than trying to outperform particular methods. In future work, we plan to investigate the relative merits of the ridge and normal intervals in greater detail.

**Table 1:** Comparison of nominal 90% confidence intervals

|  |  | oracle | ridge | normal | OLS |
|---|---|---|---|---|---|
| setting 1 | width | 0.21 | 0.20 | 0.23 | 0.16 |
| $n = 100, \; p = 45, \;\; \eta = 0.5$ | coverage | 0.90 | 0.87 | 0.91 | 0.81 |
| setting 2 | width | 0.22 | 0.26 | 0.26 | 0.06 |
| $n = 100, \; p = 95, \;\; \eta = 0.5$ | coverage | 0.90 | 0.88 | 0.88 | 0.42 |
| setting 3 | width | 0.20 | 0.21 | 0.22 | 0.16 |
| $n = 100, \; p = 45, \;\; \eta = 1$ | coverage | 0.90 | 0.90 | 0.91 | 0.81 |
| setting 4 | width | 0.21 | 0.26 | 0.23 | 0.06 |
| $n = 100, \; p = 95, \;\; \eta = 1$ | coverage | 0.90 | 0.92 | 0.87 | 0.42 |

**Acknowledgements.**  MEL thanks Prof. Peter J. Bickel for many helpful discussions, and gratefully acknowledges the DOE CSGF under grant DE-FG02-97ER25308, as well as the NSF-GRFP.

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
