[Reviews · NeurIPS 2014]

Submitted by Assigned_Reviewer_18

This paper proposes a two step procedure to apply residual bootstrap to estimate linear contrasts of c ^ T ( \hat \beta _ \ rho -\beta), where \hat \beta _ \rho is the ridge estimator. It is applicable in the regime of n > p, but p / n \to (0,1), and sufficient power law decay of the eigenvalues of the covariance \hat \Sigma. Assumption A1 rules out applying this in the p > n regime since \hat \Sigma is rank deficient. This is not a major issue, since ridge regression isn't commonly used in the p > n regime. Overall, this paper makes an interesting contribution to the study of residual bootstrap.

I have several questions
1) Would it be possible to make confidence intervals instead of prediction intervals as done in THeorem 4?
2) Is it possible to loosen A3. As p grows, the vector \norm { \beta } naturally grows.
Comments
1) Some experiments would be nice.
2) More discussion of how to choose the \rho and \scriptrho would be nice. It seems that \rho < \scriptrho. I am unclear on why \scriptrho larger helps estimating the distribution of the residuals. It seems that the value of \scriptrho only affects the bias that enters the residual , and centering should fix this.
3) Relevant work on testing linear contrasts is Exact post-selection inference with the lasso by Lee,Sun,Sun, and Taylor.
Summary: This paper is very interesting and makes an important advance in the application of residual bootstrap to the case that p / n \ to (0,1). This paper should be accepted.

Submitted by Assigned_Reviewer_19

The authors propose a theoretical analysis of the residual bootstrap for the
ridge regression estimator, under general (additive) noise model.
The analysis provided requires some structural assumption
that the empirical covariance of the variables is nearly low rank (in the sense
that its spectrum decreases polynomially fast toward zero). Under some more
technical assumptions (e.g. the design matrix is Gaussian)
the authors prove that the residual bootstrap approximates
all the conditional laws of all fitted values.

Quality: The paper is very well written and the organization is clear.
It gives new insight on the benefit of the residual bootstrap in possibly
large dimension without sparsity assumption, even when n \approx p.

Clarity: very good.

Originality: proof techniques seems rather standard, but the context is new I believe.

Significance: this is a good direction to get prediction interval in large dimension.
Such a road has still not been carefully explored until recently (apart from few papers,
that the authors mentioned).

Line by line comments:

032: requires imposes ->imposes
125: harmonize the way paper are cited (e.g. B&F 1983 is [17]!)
131: distribution distribution->distribution
133: focus on role-> focus on the role
138: Theorem ?? -> provide the correct name (potentially issue with the supplementary version one)
157: the bold notation for the conditional sign | if fairly unusual and looks weird.
167: \Phi is not defined before, only \Phi_\rho was defined in (4)
Summary of the review:
175: correct the recurring error: X^\top X+\rho : the first term is a matrix, the second one is a scalar.
The identity matrix is missing. This is everywhere in the article / supplementary.
289: a variety bounds ->a variety of bounds
310/313: can the authors give some examples where such an assumption is realistic in practice?
323:that \beta that->that \beta
413:a conclusion is missing, no? And some room is available to do so!
416: names are duplicated

Most references: please provide the full name when citing the
and double check the wrong ones like [6,8,9,10,12] where the "et al." are useless.
Note also that if you want to provide first names, do it equally for all authors (sometimes being the same):
e.g.: P. J. Bickel and Peter J. Bickel refer to the same authors!

441: citation issue again
449: idem
507/530: the constant could be tracked I guess here. Overall details are a bit loose around here.

Summary: This is an interesting paper giving theoretical insight on the residual bootstrap.
It leads to new insights on second order statistics (i.e. dealing with uncertainty quantification)
in high dimension.

Submitted by Assigned_Reviewer_40

The article generalizes the residual bootstrap to high-dimensional linear models by using ridge regression to obtain the ``pilot'' estimate of the regression coefficients. The proposed approach is original and the theoretical results are strong. The authors show using ridge regression to obtain the ``pilot'' estimate gives consistent approximations to the law of the linear contrast ($\Psi_\rho$) when the first $\min\{n,p\}$ singular values of the design matrix decay according to a power-law. The authors also show simultaneous consistency of the prediction intervals, conditioned on a random Gaussian design. Overall, the paper is well-written and the proofs are easy to follow.

My main concern is the lack of simulation results. The article will be significantly strengthened by the inclusion of some simulations on even synthetic data that, for example, show the coverage of the bootstrap confidence intervals. Some minor comments are:

1. The Mallows-$\ell_2$ metric is appropriate when analyzing the bootstrap because convergence in Mallows metric implies convergence in law and convergence of the first and second moments. Please remind readers of this fact when introducing the Mallows metric.

2. In section 2, subsection on ``a two-stage approach'', the authors use $\varrho$ to denote the regularization parameter used to compute the pilot estimate and $\rho$ to denote the regularization parameter than appear in the linear contrast. To remain consistent with the notation, most of the $\rho$'s in section 3.3 should be changed to $\varrho$'s.
Summary: The article proposes an original approach to inference for high-dimensional linear models based on the residual bootstrap. The proposed approach is original, and the theoretical results are strong, but corroborating simulation results are nonexistent.
Author Feedback
Author rebuttal: We thank the reviewers for their detailed and constructive comments.

General responses to all reviewers:

We will be happy to provide simulations to demonstrate the effectiveness of the proposed method. The simulation results have already been obtained, and we will be sure to make additional space in the paper so that they can be included.

Also, we agree with all of the technical clarifications that have been suggested, and these will be incorporated into the final version.
---------------------------------------------------

Responses to Reviewer_19:
We appreciate your detailed line-by-line comments, and we agree that these corrections should be made.
---------------------------------------------------

Responses to Reviewer_40:
We agree that simulations will be an important improvement to the paper, and we will be sure to include them in the final version. As you suggested, we will give more discussion to the properties of the Mallows metric. Lastly, we agree with your suggestion regarding the notation in Section 3.3.
---------------------------------------------------

Responses to Reviewer_18:
It is possible to allow ||beta||_2 to grow with (n,p), as long as it does not grow too quickly. We assumed ||beta||_2 = O(1) in A3 mainly for simplicity, and we will look more carefully into how much this can be relaxed.

We agree that the choice of rho and varrho merits more attention. Regarding the regularization parameter for the pilot estimator (varrho), Theorems 2 and 3 give some some guidance, but we will expand on this in the final version. (Our presentation would have been clearer if we had stated Theorem 3 using varrho instead of rho -- as suggested by Reviewer_40.) Specifically, Theorem 2 shows that varrho should minimize MSPE, and Theorem 3 gives the theoretically optimal rate, varrho ~ n^(-nu/(nu+1)), for this. In experiments, we use the theoretical value as a starting point when searching via cross validation.

The choice of rho is more problem-specific, since it depends on the application that \hat{beta}_rho is being used to serve. For estimation, rho might be chosen to minimize L2 error, whereas for prediction, rho might be chosen to minimize MSPE. To the best of our knowledge, it is not clear that there is a general relationship between rho and varrho that holds for all problems. For the particular case of prediction, your intuition that rho should be smaller than varrho is correct based on our simulations. Essentially, varrho needs to be large enough to prevent the pilot estimator from overfitting, whereas rho needs to be small enough to keep the bias term negligible in Theorem 1.

We will include the reference to Lee, Sun, Sun, and Taylor.